# Gene Expression Differences Based on Low Total 25(OH)D and Low VDBP Status with a Preterm Birth

**DOI:** 10.3390/ijms26104475

**Published:** 2025-05-08

**Authors:** Jennifer Woo, Tulip Nandu, Alexandra Nowak, Anna Forsman, Carmen Giurgescu

**Affiliations:** 1College of Nursing and Health Innovation, University of Texas Arlington, Arlington, TX 76019, USA; 2Green Center for Reproductive Biology, UTSW, Dallas, TX 75390, USA; tulip.nandu@utsouthwestern.edu; 3Marcella Niehoff School of Nursing, Loyola University Chicago, Maywood, IL 60153, USA; anowak8@luc.edu; 4Department of Biology, Colby College, Waterville, ME 04901, USA; aforsman@colby.edu; 5College of Nursing, University of Central Florida, Orlando, FL 32816, USA; carmen.giurgescu@ucf.edu

**Keywords:** preterm birth, pregnancy, gene expression, transcriptome, vitamin D, vitamin D deficiency, 25-hydroxyvitamin D 2, vitamin D-binding protein

## Abstract

Preterm birth (PTB; <37 weeks’ gestation) is a persistent problem in the United States that affects non-Hispanic Black women at much higher rates than White women. Several biomarkers have been associated with PTB, including vitamin D deficiency (VDD) and low levels of vitamin D-binding protein (VDBP). However, no biomarker has been found to predict PTB. To identify a predictive biomarker of PTB, gene expression differences were determined in Black women with PTB and full-term births and between women with high and low levels of plasma vitamin D and high and low VDBP levels. In this pilot study of 19 pregnant women from the Biosocial Impact on Black Births (BIBB) study, we found that 47 genes were upregulated and 16 genes were downregulated in women with PTB as compared with women who had a full-term birth, 361 genes were downregulated and 61 genes were upregulated in women with VDD as compared with those that had vitamin D sufficiency, and 44 genes were upregulated and 295 were downregulated in women with low VDBP. Several genes expressed by neutrophils were downregulated in the PTB, VDD, and low VDBP groups. These findings support the idea that vitamin D and VDBP status may be important clinical markers influencing the gene expression of genes associated with PTB.

## 1. Introduction

In the United States, the incidence of preterm birth (PTB) (born < 37 completed weeks’ gestation) is higher than in other developed, high-income nations [1]. This incidence is also higher in non-Hispanic Black women (14.58%) than in non-Hispanic White women (9.44%) [2]. Yet, interventions to mitigate PTB risk, such as 17-hydroxyprogesterone acetate, have not been effective [2,3]. The disproportionately high incidence of PTB in Black women is not associated with lower socioeconomic status, insurance type, or maternal education, as there is evidence to show that Black women of high socioeconomic status with unlimited access to healthcare are still at higher risk for PTB when compared to non-Hispanic White women [4]. Typical social determinants of health do not explain the disparity in PTB outcomes, so it is important to understand the gene and environmental influences that are impacting PTB risk in Black women, such as exposure to perinatal stress and racism and how these factors may impact inflammation and PTB risk [5,6]. PTB is the leading cause of neonatal mortality and morbidity; thus, it not only impacts the offspring’s immediate health, but it may influence their future health as adults through epigenetic changes [2,5,7].

One challenge in reducing PTB risk is that PTB can be influenced by many environmental, maternal, genomic, and epigenomic factors [8]. Approximately 40% of PTBs are idiopathic and occur spontaneously due to preterm labor, 25% to 35% are due to the premature rupture of membranes, and 30% to 35% are due to medically indicated causes of PTB, such as pre-eclampsia [9]. In many cases, the cause of PTB is unknown. In a cross-country analysis of 4.1 million singleton births in five high-income countries, 65% of PTBs could not be explained by any plausible biologic explanation or known risk factor [9,10].

One strategy to mitigate PTB risk is to identify genetic biomarkers that accurately predict PTB. In a multi-ethnic group of women, researchers found transcriptomic differences in the placenta of women who had a PTB due to acute histologic chorioamnionitis vs. idiopathic spontaneous PTB [11]. However, this invasive approach of identifying transcriptomic differences in placentas after birth is not feasible in identifying high-risk women prenatally. A safer approach would be to identify a biomarker that can be detected early in pregnancy and in samples routinely collected from pregnant women so that interventions can be undertaken to prevent PTB.

Black women are at a higher risk of vitamin D deficiency (VDD; defined as 25(OH)D <20 ng/mL or <50 nmol/L) because of increased melanin, which deflects ultraviolet rays [12,13]. Pregnancy is also a risk factor for VDD [14]. VDD has been related to PTB [15,16]. However, the predictive power of VDD has not been consistent across different populations [17]. 25-hydroxyvitamin D [25(OH)D] has an important immunomodulatory role in pregnancy that could protect against infection and inflammation associated with PTB [18,19]. Several pregnancy-related tissues also express a vitamin D receptor. These tissues include all immune cells and the cervix, uterus, and vagina, which could all play a role in PTB [20,21,22,23]. The molecular mechanisms of this effect have been well studied in vitro. For example, calcitriol (also known as 1,25-hydroxyvitamin D) exerts anti-inflammatory effects on peripheral blood mononuclear cells (PBMCs), including monocytes, macrophages, T cells, and lymphocytes [24]. In a systematic review of vitamin D and inflammatory status, increased calcitriol consistently downregulated interleukin-6 (IL-6), tumor necrosis factor-α (TNF-α), IL-1β, and interferon-γ [24]. Calcitriol also upregulated IL-10, an important anti-inflammatory cytokine [24]. Many of these inflammatory cytokines have been implicated in PTB, and specifically in Black women who are pregnant [25,26]. Thus, 25(OH)D status may influence the gene expression of immune-related functions linked to PTB.

Another factor associated with PTB and linked to VDD is vitamin D-binding protein (VDBP). VDBP is the principal transporter of 25(OH)D and is important for the entry of 25(OH)D into renal tubular cells, where 25(OH)D is converted into 1,25-hydroxyvitamin D [27]. Thus, VDBP may modulate biological functions, such as those related to immunomodulatory actions in pregnancy [28]. Unfortunately, clinicians do not have an established cutoff for VDBP concentrations in pregnancy; thus, these data are not currently collected except in research studies. The evidence for the role of VDBP in PTB is mixed, but one hypothesis is that higher levels of VDBP decrease the amount of calcitriol entering cells to exert biological activities, especially those related to immune function [29]. Interestingly, low VDBP amounts in the first trimester of pregnancy were predictive of spontaneous PTB among White and East Asian women [30]. However, in Korean women, higher VDBP amounts in serum collected during the third trimester of pregnancy were associated with PTB, maternal age, parity, and body mass index [31]. Although these results show variability between different populations of women and stages of pregnancy, they link VDBP levels to PTB risk and could help to uncover a genetic biomarker of PTB. No studies to date have examined the association of VDBP levels with the risk of PTB among a cohort of Black women.

In this pilot study, we aimed to identify potential genetic biomarkers of PTB using a comprehensive RNAseq approach. To this end, we compared differences in gene expression patterns associated with plasma 25(OH)D and VDBP levels in a cohort of Black women who had either a PTB or full-term birth.

## 2. Results

Most participants had a relatively low socioeconomic status and were on Medicaid for health insurance coverage. Approximately one-third of women were married. Independent samples *t*-tests for were used for continuous variables and Fisher’s exact tests (due to the small sample size) were used for categorical variables to test for differences in the preterm vs. term groups, VDD and vitamin D-sufficient groups, and the low VDBP vs. high VDBP groups. We found no statistically significant differences in various health and sociodemographic characteristics between women who had PTB (<37 weeks’ gestation; n = 13) and women who had full-term births (≥37 weeks’ gestation; n = 6) (Table 1). However, women in the PTB group had greater incidence of hypertensive disorders of pregnancy than those in the full-term group. We also found no statistically significant differences in these characteristics between women with VDD (<20 ng/mL; n = 12) and women with vitamin D sufficiency (≥20 ng/mL; n = 7). Although clinical cutoffs have not been established for VDBP, the amounts of VDBP can be analyzed for research [31]. In this study, we classified low VDBP as levels lower than or equal to 510 mcg/mL and high VDBP as levels greater than 510 mcg/mL since 510 mcg/mL was the median in the sample. Finally, there were no statistically significant differences in these characteristics between women with low levels of VDBP (n = 13) and women with high levels of VDBP (n = 6). We also found no significant differences in gestational age at blood draw between the groups since 25(OH)D levels change across pregnancy.

### 2.1. Differential Gene Expression Analysis Based on Preterm vs. Full-Term Births

We evaluated differences in gene expression between women who had PTB and women who had full-term births. The analysis revealed 47 upregulated genes and 16 downregulated genes in women with PTB vs. women with full-term births. The top 10 upregulated genes and top 10 downregulated genes are indicated in Table 2; the complete list is included in Appendix A. These genes were not significantly differentiated based on *Q* values. However, these genes were differentially expressed when using a less-conservative cutoff (*p* < 0.05).

### 2.2. Differential Gene Expression Analysis Based on Vitamin D Status

We also evaluated differences in gene expression between participants with vitamin D deficiency (<20 ng/mL) and participants with vitamin D sufficiency (≥20 ng/mL). This analysis revealed 361 downregulated genes with a fold change of −0.67 and 63 upregulated genes with a fold change >1.5 in the vitamin D-deficient group vs. the vitamin D-sufficient group. The top 10 upregulated and downregulated genes based on log fold change are listed in Table 3. Of these genes, two were downregulated in both women with a PTB and women with VDD: *OLFM4* and *DEFA4*.

### 2.3. Differential Gene Expression Analysis Based on Vitamin D-Binding Protein Status

Although clinical cutoffs have not been established for VDBP, the amounts of VDBP can be analyzed for research [31]. In this study, we classified low VDBP levels as ≤510 mcg/mL and high VDBP levels as >510 mcg/mL because 510 mcg/mL was the median concentration of VDBP across the samples. Differential gene expression analysis revealed 44 upregulated and 295 downregulated genes in women with low VDBP levels vs. women with high VDBP levels. The top 10 of these upregulated and downregulated genes based on log fold change are listed in Table 4.

### 2.4. Intersection of Differential Gene Expression Analysis Among Groups

We evaluated whether all three groups—PTB, VDD, and high VDBP—shared any differentially expressed genes. There were 2 genes upregulated in all three groups and 10 genes downregulated in all three groups (Figure 1; complete list in Appendix A). We assessed fold-change differences in gene expression among the PTB, VDD, and high VDBP groups with volcano plots (Figure 2).

#### Gene Ontology Based on DGE

Based on the 10 differentially expressed genes downregulated in all three groups, PTB, VDD, and low VDBP, which are listed in the attached Appendix A, are genes expressed by neutrophils (OLFM4, TCN1, RETN, PGM5, RNASE3, ORM1, BPI, MMP8, DEF4A, and CHIT1). Only two genes were upregulated in all three groups (FCLR5 and TDRD9) Table 5 shows functional annotation clustering of the significantly altered pathways involved in the genes of interest using the David gene ontology tool (https://david.ncifcrf.gov/conversion.jsp?VFROM=NA; accessed on 10 September 2024). The gene ontology terms involved in these genes that were downregulated could be potential targets for upregulation with increased calcitriol that could result from vitamin D supplementation.

## 3. Discussion

In this study, we aimed to identify gene expression patterns associated with VDD and low VDBP levels that could be predictive biomarkers of PTB in a cohort of Black women. The 10 genes that were downregulated in all three groups (PTB, VDD, and low VDBP) are involved in innate immune responses and the modulation of inflammatory pathways related to IL-6 and TNF*α*. In addition, the FCRL5 and TDRD9 genes were upregulated in all three groups. These findings suggest that 25(OH)D and VDBP may have important influences on the gene expression of inflammatory-related genes early in pregnancy that could potentially impact PTB risk later in pregnancy.

We found that OLFM4 and DEFA4 were both downregulated in the VDD, low VDBP, and PTB groups. Both of these genes are involved in initiating and maintaining the immune response. OLFM4 encodes a protein that is (1) an antiapoptotic factor that promotes tumor growth and (2) an extracellular matrix glycoprotein that facilitates cell adhesion and has been predominantly associated with gastric/intestinal function [33,34]. OLFM4 bolsters innate immunity by regulating pathogen-responsive inflammation and mitigating inflammation and tissue injury [33]. Although these pathways are associated with bacterial infection, gastrointestinal inflammation, and cancer, they suggest that OLFM4 may be protective of PTB as it relates to cervical epithelium [35]. Indeed, much research on OLFM4 has been related to the intestine and its cell-protective role in the inflamed mucosa of irritable bowel disease [36,37,38]. However, similar findings could arise from research on the cervix and its role in inflammation-mediated PTB caused by infection or the degradation of cervical integrity but could potentially translate to neutrophils in the blood.

Another neutrophil-expressed gene downregulated in the VDD, low VDBP, and PTB groups was DEFA4, which encodes a family of antimicrobial and cytotoxic peptides involved in host defense [39]. Though there is no published study that directly ties DEF4A with PTB, other studies have found that downregulated genes of antimicrobial peptides in the same family as DEFB1 have been associated with PTB [40]. In a cohort study of 14 women who had full-term births, OLFM4 and DEFA4 were upregulated in PBMCs during the first trimester and early second trimester of pregnancy when compared with the pre-pregnancy time period and the third trimester and postpartum period. The upregulation of neutrophil-dominant genes early in pregnancy provides some evidence that neutrophil-expressed genes may have played an important immunomodulatory role early in pregnancy in a cohort of Danish pregnant women with a normal full-term birth [41]. Since our cohort included only Black/African American pregnant women, it is possible that the changes seen in our study could be related to race/ancestry, but this needs to be fully explored in a multi-ethnic cohort.

Another study also linked immune-related gene expression signatures to VDD and PTB [42]. In this study, PBMCs were isolated from blood draws collected from a multi-ethnic cohort of pregnant women between 8 and 23 weeks of gestation (similar to our study). These PBMCs underwent transcriptomic analysis, which correlated VDD and PTB to gene signatures related to maternal systemic changes in immune (innate and adaptive) and inflammatory processes. Specifically, the authors found that women who had both spontaneous PTB and VDD had increased connectivity of expression of these genes, OLFM4, DEF4A, IL-10, IL-6, and IL-8, suggesting potentially linking vitamin D to the modulation of innate and adaptive immune responses [42].

In our study, we also found that EDAR was upregulated in the VDD group. EDAR is a member of the tumor necrosis factor family of genes that can activate the nuclear factor-kappa beta (NF-κβ) pathway [43]. In several studies, the NF-κβ pathway has been associated with PTB in both humans and animal models due to its role in promoting the onset of labor [44,45,46,47]. Calcitriol has been shown in murine and in vivo models to downregulate the NF-κβ pathway; however, no studies have been conducted in pregnancy models [48,49]. Calcitriol may have an important role in downregulating pro-inflammatory-related genes such as EDAR and IL-6, which can increase the risk of PTB [50].

Further, we identified other genes that may be associated with PTB, including CHIT1, PRTN3, and MMP8. CHIT1 is expressed in activated human macrophages, which are important for the immune response to potential infections [51]. PRTN3 was higher in the amniotic fluid of women with an imminent PTB compared to women who had a full-term birth, as evidenced in a predominantly African American cohort of 90 pregnant women with a diagnosed short cervix; however, maternal blood was not assessed [52]. MMP8 is an enzyme neutrophil collagenase that has also been implicated in intra-amniotic infection and PTB [53]. MMP-8 and other members of the matrix metalloendopeptidase family have been associated with greater risk for premature rupture of membranes [52,53]. MMP-8 in maternal blood was downregulated in women with PTB and VDD [42,54], which aligns with our findings in the PTB and VDD groups. Although we do not know the significance of these factors in PBMCs, our findings suggest that early downregulation of CHIT1, MMP8, and PRTN3 expression may be a precursor for increased risk of PTB. If indeed these genes are important for the immunosenescence of pregnancy, could vitamin D supplementation upregulate these genes and potentially improve birth outcomes?

Based on the number of genes differentially expressed in the low and high VDBP groups, our data suggest that VDBP may influence the relationship between vitamin D status and immunological function in pregnancy [28,55,56]. VDBP status is highly influenced by estrogen during pregnancy and could modulate bioavailable vitamin D [57]. VDBP concentrations were found to be an important marker for bioavailable vitamin D, regardless of the total 25(OH)D status in healthy patients, patients in the intensive care unit, and Korean women who were pregnant [58,59]. Thus, depending on the degree of VDBP downregulation, significant changes in bioavailable vitamin D could lead to negative health outcomes.

In addition, the two genes that were upregulated in the PTB, VDD, and low VDBP groups have been identified as biomarkers for PTB risk in other studies [60,61]. Upregulation of the gene FCRL5 has been associated with autoimmune disease, which may be an important factor in pregnancy in the context of anti-fetal rejection, and the question remains of whether increasing vitamin D status could mitigate the upregulation of this gene [62,63].

Another important factor to consider is VDBP polymorphisms. For example, the GC allele of VDBP has approximately 120 unique single-nucleotide polymorphisms (SNPs) that profoundly impact vitamin D metabolism [31]. In people of African ancestry, the most common VDBP polymorphism is the homozygous 1F allele genotype (rs7041-T; rs4588-C) [64,65]. People with this genotype need higher vitamin D supplementation to raise their total 25(OH)D concentrations [65,66]. There is evidence that specific VDBP SNPs are related to PTB and gestational diabetes mellitus [31,67]. There is also evidence that vitamin D supplementation can improve PTB outcomes in populations with severe VDD [15,68,69]. These findings suggest that the interaction between genetic factors, such as VDBP polymorphisms and environmental factors, such as exposure to racism and perinatal stress [70,71], can impact outcomes such as PTB. In a large cohort of pregnant Black women, early-pregnancy vitamin D deficiency was associated with a two-times greater risk of PTB [15]. Therefore, more research is needed to assess the potential of vitamin D supplementation as an intervention for decreasing the risk of PTB.

The next step is to understand the genetic underpinnings of vitamin D metabolism by investigating genes such as VDBP or Vitamin D Receptor (VDR) that could be related to vitamin D supplementation in pregnancy at the single-cell level. This approach will help to further elucidate how calcitriol impacts leukocytes, such as natural killer cells, T cells, macrophages, and neutrophils, at various points during pregnancy. These findings could provide theoretical and physiological evidence for mitigating PTB risk in pregnant women, especially in Black women who are more at risk for having VDD.

One limitation of this study is the small sample size. This study was a pilot to examine differences based on gestational age at birth, 25(OH)D status, and VDBP status in a cohort of Black pregnant women in the late first trimester and mid-second trimester. Because gene expression changes across pregnancy, having access to samples collected at a single timepoint is also a limitation. Analyzing plasma 25(OH)D at multiple timepoints in pregnancy could have strengthened our findings and the study design, but due to grant constraints, the analysis of plasma 25(OH)D was only included at the first timepoint. To strengthen the results found in this study, it should be validated in another cohort of pregnant women to see if we can replicate our findings. In addition, because this was a retrospective secondary data analysis, we did not have dietary information on the participants since it was not part of the original parent study; therefore, we could not control for dietary vitamin D and calcium intake and/or make an accurate assessment of whether the participants were taking any vitamin D supplementation. However, our results provide an important foundation for future work to be conducted, as our study relates to understanding the molecular mechanism by which calcitriol can make an impact immune cell type in pregnancy.

## 4. Materials and Methods

### 4.1. Patients

This retrospective case–control, pilot study evaluated data from a subsample of the Biosocial Impact on Black Births (BIBB) study. The BIBB study was a prospective, longitudinal study that examined biopsychosocial aspects of PTB in Black women, which occurred from 2017 to 2023. Women were enrolled at varying timepoints in the parent study (8 to 29 weeks gestation), were 18 to 45 years old, had a singleton pregnancy, were able to speak and read English, and lived in Michigan or Ohio. However, for the participants included in this pilot study, only participants who had blood drawn between 8 and 24 weeks of gestation were included. Because the BIBB study examined biological markers (e.g., cytokines), women were excluded if they had autoimmune disease, were taking anti-inflammatory medications, or had a fever at the time of data collection. Enrolled women completed surveys and had blood drawn at three timepoints during their pregnancies. For our study, we selected a convenience sample of 19 women from the BIBB study who were recruited between 8 and 19 weeks of gestation. The blood samples collected at the first timepoint were used for these analyses as a secondary data analysis. We included 13 women who experienced spontaneous PTB, defined as premature labor or preterm, premature rupture of membranes, and six women with full-term births matched by maternal age and gestational age at data collection with women with PTB.

In the BIBB study, trained research assistants screened women for eligibility during their prenatal visits. All participants completed an informed consent process. During their prenatal visits, participants completed surveys on computer tablets equipped with QualtricsXM data collection software (Qualtrics International, Inc., Provo, UT, USA) (https://www.qualtrics.com/) and had venous blood drawn. The survey included questions to collect data on maternal age, marital status, employment, annual household income, and smoking during pregnancy. Gestational age at enrollment, gestational age at birth, and maternal medical history including history of PTB in a prior pregnancy were extracted from electronic health records.

The BIBB study was approved by the Institutional Review Board at Wayne State University and Ohio State University; this substudy was considered by the Institutional Review Board at Texas Woman’s University due to use of de-identified data.

### 4.2. Sample Collection

Peripheral blood was collected in sterile EDTA-treated blood-collection tubes and in RNA PaxGENE^®^ blood-collection tubes (PreAnalytiX Qiagen/BD, Hombrechtikon, Switzerland) to preserve sample integrity for later analysis. Blood secured in the EDTA preservative was centrifuged, and plasma was aliquoted into 1.5 mL collection tubes. RNA PaxGENE^®^ blood tubes were processed according to the manufacturer’s instructions. Both samples were stored at −80 °C until analysis.

### 4.3. Plasma 25(OH)D Analysis

Samples were extracted and analyzed by Heartland Assays. All standards, controls, and samples were analyzed by tandem liquid chromatography–mass spectrometry (LC/MS) using an Agilent 1290 Infinity (Agilent, Santa Clara, CA, USA), high-performance liquid chromatography system coupled to an Agilent 6460 MS/MS (Agilent, Santa Clara, CA, USA), with an electrospray ionization source. Assay accuracy is greater than 95% based on the standard assessment for 25(OH)D certified by the National Institute of Standards and Technology. Controls for 25(OH)D in plasma are greater than 90% accurate and have an inter-assay and intra-assay coefficient of variation less than 5%.

### 4.4. Vitamin D-Binding Protein Analysis

Plasma samples were extracted and analyzed by Heartland Assays. All samples were analyzed using a double-antibody sandwich ELISA that is highly sensitive for measuring VDBP. An Aviva Systems Biology VDBP ELISA kit (OKIA00086) (Aviva Systems Biology Corp., San Diego, CA, USA), was used for VDBP analysis. A pre-coated 96-well plate with VDBP antibody with standards or test samples added to the wells was used, then incubated and washed to remove unbound proteins. Then, an anti-VDBP-HRP conjugated detector antibody was added. The complete protocol can be found at Aviviasysbio.com (https://www.avivasysbio.com/vitamin-d-binding-protein-elisa-kit-human-okia00086.html; accessed on 10 January 2023).

### 4.5. RNA Extraction and Library Creation

RNA isolation was from blood collected in RNA PAXgene^®^ tubes and has been described in detail in Nowak et al.’s study [72]. Utilizing a MagMax™ RNA isolation kit (TheromFisher Scientific, Waltham, MA, USA), samples were converted to complementary DNA and were shipped to the University of Central Florida Genomics and Bioinformatics facility for quality control and library creation.

Libraries were prepared from total RNA using the NEBNext Ultra II Directional RNA Library Prep Kit (Illumina, San Diego, CA, USA) and were sequenced by Genewiz (Illumina HiSeq, 2 × 150 bp). The raw data underwent quality control analyses using the FastQC (version 0.11.9) tool. Reads were then mapped to the human genome (hg38) using the spliced read aligner TopHat (version 2.0.13). Transcriptome assembly was performed using Cufflinks (version 2.2.1) with default parameters. The transcripts were merged into distinct, non-overlapping sets using cuffmerge, followed by cuffdiff to call the differentially expressed transcripts. The FPKM (fragments per kilobase of transcript per million mapped reads) expression values from the cuffdiff analysis were used to identify changes in gene expression.

## 5. Conclusions

This pilot study provides some evidence that vitamin D may play an important immunomodulatory role in mitigating the risk for PTB. Neutrophil-dominated genes were downregulated in the PTB, VDD, and low VDBP groups. It is still unclear what impact VDBP status has on PTB risk, but this warrants further investigation. Due to the inconsistencies that have been documented in the literature related to vitamin D supplementation, future studies must examine the exact pathways that vitamin D can impact from a molecular perspective to maximize treatment and improve outcomes.

## Figures and Tables

**Figure 1 ijms-26-04475-f001:**
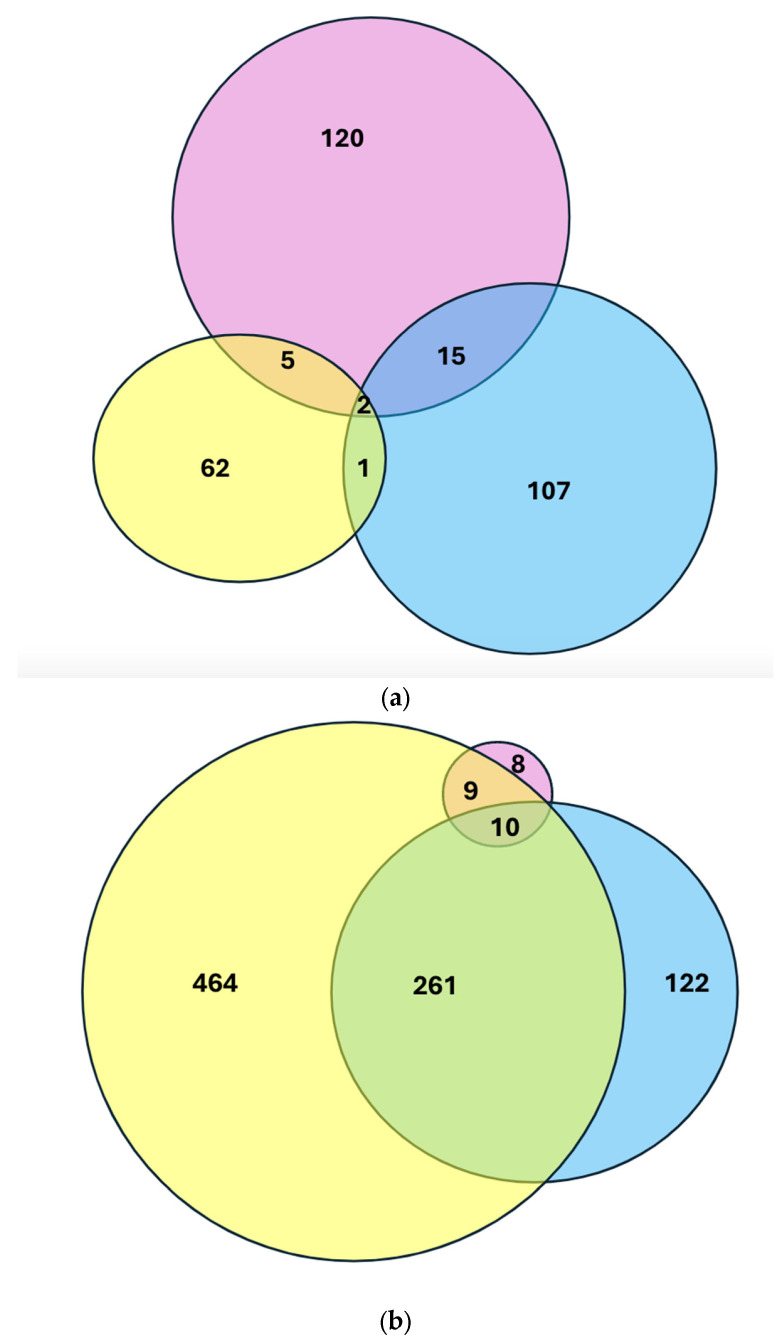
Overlap of differential gene expression analysis in PTB (pink), VDD (yellow), and low VDBP (blue) groups. (**a**) Genes upregulated in the PTB group (pink), VDD group (yellow), and low VDBP group (blue). (**b**) Genes downregulated in the PTB group (pink), VDD group (yellow), and low VDBP group (blue). PTB, preterm birth; VDD, vitamin D deficient; VDBP, vitamin D-binding protein [32].

**Figure 2 ijms-26-04475-f002:**
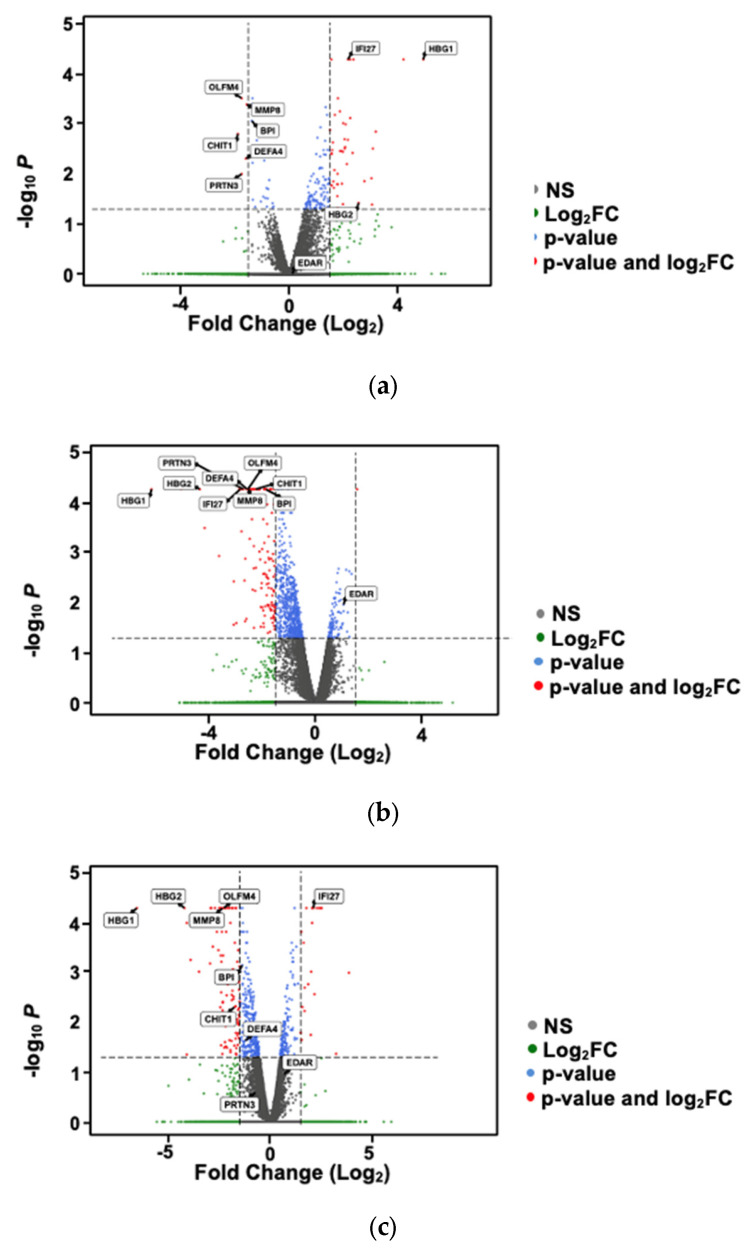
Volcano plots of fold-change differences in gene expression based on the PTB group (**a**), VDD group (**b**), and low VDBP group (**c**). FC, fold change; NS, not significant, PTB, preterm birth; VDBP, vitamin D-binding protein; VDD, vitamin D deficient. Dotted lines represent 1.5-fold change marker.

**Table 1 ijms-26-04475-t001:** Characteristics of the sample population (n = 19).

Characteristics	Preterm Birth (n = 13)	Full-Term Birth (n = 6)	Vitamin D Deficiency (n = 12)	Vitamin D Sufficiency (n = 7)	High VDBP (n = 6)	Low VDBP (n = 13)
Maternal age, years, M ± SD	27.54 ± 6.2	30.17 ± 7.19	28.17 ± 5.9	28.7 ± 7.8	27.2 ± 6.4	28.9 ± 6.6
Gestational age at blood draw, weeks, M ± SD	14.92 ± 3.79	11.33 ± 3.07	12.83 ± 3.09	15.3 ± 4.79	12.0 ± 2.1	14.6 ± 4.3
Level of education, N (%)						
Less than high school	3 (23.1)	1 (16.6)	2 (16.6)	2 (28.6)	1 (16.6)	3 (23)
High school or equivalent	4 (30.7)	2 (33.3)	5 (41.7)	1 (14.3)	2 (33.3)	4 (30.7)
> High school education	5 (38.4)	3 (50)	5 (41.7)	4 (57.1)	3 (50)	6 (46.3)
Marital status, N (%)						
Married/lives with partner	4 (30.7)	3 (50)	4 (33.3)	3 (42.8)	2 (33.3)	5 (38.4)
Work status N (%)						
Currently working	6 (38.5)	3 (50)	6 (50)	3 (42.8)	3 (50)	7 (54)
Health insurance N (%)						
Medicaid *	7 (53.8)	3 (50)	5 (41.6)	6 (85.7)	2 (33.3)	8 (61.7)
Medicare	3 (23.1)	1 (16.6)	4 (33.3)	0 (0)	2 (33.3)	1 (7.6)
Medicaid + Medicare	1 (7.6)	0 (0)	1 (8.3)	0 (0)	0 (0)	1(7.6)
Private or other	1 (7.6)	1 (16.6)	0 (0)	1 (14.3)	2 (33.3)	3 (23.1)
HDP diagnosis ** N (%)	7 (53.8)	1 (16.6)	3 (25)	4 (57.1)	4 (66.6)	3 (23.1)
Obese (BMI ≥ 30 kg/m^2^), N (%)	7 (53.8)	3 (50)	4 (33.3)	3 (44.6)	3 (50)	7 (53.8)

Abbreviations: BMI, body mass index; HDP, hypertensive disorders in pregnancy. * Medicaid is defined as health insurance coverage provided by federal/state agencies for low-income individuals. ** statistically significantly higher HDP diagnosis in PTB vs full-term birth group *p* < 0.01.

**Table 2 ijms-26-04475-t002:** Upregulated and downregulated genes in the preterm birth group.

Log2 Fold-Change Up ^1^	Upregulated Genes	Log2 Fold-Change Down ^1^	Downregulated Genes
4.93	*HBG1*	−1.87	*CHIT1*
4.19	*SERINC2*	−1.73	*OLFM4*
3.18	*MAN1A2*	−1.72	*PRTN3*
3.08	*LY96*	−1.58	*DEFA4*
3.05	*LINC01506*	−1.55	*MMP8*
3.02	*RSAD2*	−1.35	*BPI*
2.76	*CLEC2B*	−1.34	*TMEM180*
2.55	*HBG2*	−1.33	*ORM1*
2.36	*RPS24*	−1.32	*RNASE3*
2.33	*RPL31*	−1.31	*CTSG*

^1^ *p* < 0.05.

**Table 3 ijms-26-04475-t003:** Upregulated and downregulated genes in the vitamin D-deficient group.

Log2 Fold-Change Up ^1^	Upregulated Genes	Log2 Fold-Change Down ^1^	Downregulated Genes
1.56	*FCRL5*	−6.128	*HBG1*
1.35	*LYPD2*	−5.02	*SERINC2*
1.263	*IL12RB2*	−4.32	*HBG2*
1.260	*S100A8*	−2.85	*ABCC13*
1.20	*PACSIN1*	−2.82	*IFI27*
1.18	*MZB1*	−2.80	*S100A8*
1.16	*RRM2*	−2.76	*HRAT92*
1.14	*TDRD9*	−2.68	*PRTN3*
1.06	*EDAR*	−2.54	*DEFA4*
1.05	*FBLN2*	−2.53	*OLFM4*

^1^ *Q* value < 0.05.

**Table 4 ijms-26-04475-t004:** Upregulated and downregulated genes in the low vitamin D-binding protein group.

Log2 Fold-Change Up ^1^	Upregulated Genes	Log2 Fold-Change Down ^1^	Downregulated Genes
3.88	*AP2B2*	−6.55	*HBG1*
3.25	*RSAD2*	−4.21	*HBG2*
2.544	*CMPK2*	−4.10	*SERINC2*
2.49	*SIGLEC1*	−2.92	*COX7B*
2.41	*HERC5*	−2.72	*RPS24*
2.29	*IFI44L*	−2.49	*CD177*
2.18	*IFIT3*	−2.44	*CCDC34*
2.12	*IFI27*	−2.40	*DNAJC25*, *DNAJC25-GNG10*
2.069	*OASL*	−2.32	*MMP8*
2.068	*USP18*	−2.28	*OLFM4*

^1^ *Q* value < 0.05.

**Table 5 ijms-26-04475-t005:** Biological process pathways involving downregulated DGE in PTB, VDD, and low VDBP groups.

Gene Ontology Term	Related Genes	*p*-Value	FDR (Benjamini–Hochberg)
Defense Response to Gram-negative bacterium (GO:0050829)	*BPI*, *DEFA4*, and *RNASE3*	0.00081	0.044
Innate Immune Response in Mucosa (GO:0002227)	*DEFA4* and *RNASE3*	0.017	0.32
Innate immune response (GO:0045087)	*BPI, DEFA4*, and *RNASE3*	0.023	0.32
Negative regulation of interleukin-6 production (GO:0032715)	*BPI* and *ORM1*	0.031	0.32
Antibacterial humoral response (GO:0019731)	*DEFA4* and *RNASE3*	0.034	0.32
Negative Regulation of Tumor Necrosis factor production (GO: 0032720)	*BPI* and *ORM1*	0.035	0.32
Antimicrobial humoral immune response mediated by antimicrobial peptide (GO:0061844)	*DEFA4* and *RNASE3*	0.049	0.34
Positive Regulation of tumor necrosis factors production (GO:0032760)	*MMP8* and *ORM1*	0.050	0.34
Defense response to Gram-positive bacterium (GO:0050830)	*DEFA4* and *RNASE3*	0.058	0.35
Carbohydrate metabolic process (GO:0005975)	*CHIT1* and *PGM5*	0.067	0.36
Cellular Response to Lipopolysaccharide (GO:0071222)	*DEFA4* and *MMP8*	0.08	0.41

## Data Availability

Data will be made available upon reasonable request to the coauthors.

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
