# Peer review of "Gene Expression Differences Based on Low Total 25(OH)D and Low VDBP Status with a Preterm Birth"

_ijms, 2025, doi:10.3390/ijms26104475_

Round 1
Reviewer 1 Report
Comments and Suggestions for Authors
This manuscript reports the results of a study inquiring on the possible role of Vit D in pre-term Birth (PTB). Deficiency in Vit. D (VDD) and vit. D binding protein (VDBP) have been correlated with PTB, but the exact nature of this correlation is still largely undefined. For this retrospective case-control, pilot study, data from a subsample of the Biosocial Impact on Black Births (BIBB) study were used. The BIBB study is a prospective, longitudinal study that examined biopsychosocial aspects of PTB in Black women, from 2017-2023. Women of any gestation were included if they self-identified as Black or African American, were 8 to 29 weeks gestation, were 18 to 45 years old, had a singleton pregnancy, were able to speak and read English, and lived in 312 Michigan or Ohio. In particular, the authors selected a convenience sample of 19 women from the BIBB study, all between 8-19 weeks gestation. The blood samples collected at the first timepoint were used for the analysis. The study included 13 women who experienced spontaneous PTB (defined as premature labor or preterm, premature rupture of membranes), and 6 women with full term births matched by maternal age and gestational age at data collection with women with PTB (verbatim). The authors assessed gene upregulated or downregulated in women with VDD and not, and with VDBP deficiency and not. The obtained results indicate that 47 genes were upregulated and 16 genes were downregulated in the majority of the women with PTB as compared with women who had a full-term birth, 361 genes were downregulated and 61 genes were upregulated in women with VDD as compared with those that had vitamin 19 D sufficiency, and 44 genes were upregulated and 295 were downregulated in women with low 20 VDBP. Several genes expressed by neutrophils but also in epithelia were downregulated in the PTB, VDD, and low 21 VDBP groups. These findings support the idea that vitamin D and VDBP status may be important 22 clinical markers influencing gene expression of genes associated with PTB.
Overall, The study appears to be properly conceived and conducted, and the results reported - although correlative - pave the road to future more mechanistic studies to determine the actual relevance of the identified genes in PTB. The inclusion criteria appear appropriate.
Comments: A couple of points need further clarification.
1. It would appear that for the study the authors used blood samples withdrawn only at the beginning of the study. Why no another sample (or other samples ) were withdrawn during the progression of the gestation (e.g. once per trimester or at least at the end of the gestation (either PTB or at-term) to have a better time line of both VDD or VDBP deficiency?
2. Was any attempt made to determine whether the deficiency in Vit. D and VDBP were possibly associated with changes (or lack of changes) in the diet?
3. In lieu of some discrepancies in pregnant women of other races reported and references by the authors in the manuscript, how can it be excluded that the observed gene up-regulations and down-regulations are not strictly race related?
Author Response
My rebuttal is included in the attached word document. Responses to each reviewer is included in the word document.

Reviewer 2 Report
Comments and Suggestions for Authors
Preterm birth (PTB) is a persistent problem, especially affects non-Hispanic Black women and is associated with vitamin D deficiency (VDD) and low levels of vitamin D-binding protein (VDBP).
They try to identify gene expression patterns associated with VDD and low VDBP levels that could be predictive biomarkers of PTB in a cohort of Black women.
They looked for which genes were upregulated and downregulated in three groups:
1) between women who had PTB (n = 13) and women who had full-term births (n = 6).
2) women with VDD (n=12) and women with vitamin D-sufficiency (n=7).
3) between women with low levels of VDBP (n=13) and women with high levels of VDBP
They found several genes expressed by neutrophils were downregulated in the PTB, VDD, and low VDBP groups. These findings support the idea that vitamin D and VDBP status may be important clinical markers influencing gene expression of genes associated with PTB.
There were 2 genes upregulated in all three groups (FCLR5, TDRD9) and 10 genes downregulated in all three groups (OLFM4, TCN1, RETN, PGM5, RNASE3, ORM1, BPI, MMP8, DEF4A, CHIT1)
These genes are involved in innate immune responses and modulation of inflammatory pathways related to IL-6 and TNF.
These findings suggest that 25(OH)D and VDBP may have important influences on gene expression of inflammatory-related genes early in pregnancy that could potentially impact PTB risk later in pregnancy.
Based on the number of genes differentially expressed in the low and high VDBP groups, data suggests that VDBP may influence the relationship between vitamin D status and immunological function in pregnancy.
I think the article is excellent. My point of view is that of a doctor who is an expert in Vitamin D, but fundamentally I am a doctor who treats patients, does clinical research and writes papers. I have published 13 works about Vitamin D in the last 6 years. But I'm not an expert in genetics.
The idea is excellent, the work design is very good, and the results are very interesting. There are publications that have shown discordant results in other studies on Vitamin D supplementation in different health outcomes. Without a doubt, genetics can explain these discrepancies in some cases, since beneficial effects have been documented in the presence of certain polymorphisms of several genes.
The idea of studying the expression of genes and choosing groups related to the pathology under study (VDD and low VDBP for Preterm birth is excellent. And they found what they were looking for. These genes are related to different aspects of the immune response. I have no objections and I congratulate the authors.
Author Response
Responses to the reviewers comments are in the attached word document.

Reviewer 3 Report
Comments and Suggestions for Authors
The study investigates gene expression differences in Black women with preterm births (PTB) and full-term births, based on levels of vitamin D (25(OH)D) and vitamin D-binding protein (VDBP). The results suggest that the status of vitamin D and VDBP can influence the expression of genes associated with PTB.
There are a few things that should be improved or clarified:
- Explanation of Medicaid: It should be explained what "Medicaid for health insurance coverage" means.
- Clarification of inclusion criteria: In line 28, you explained that a preterm birth is <37 weeks, but in the materials and methods section, you wrote "Women of any gestation were included if they self-identified as Black or African American, were 8 to 29 weeks gestation". Why did you choose women who gave birth between 8-29 weeks? Why not between 8-37 weeks?
- Clarification of the study sample: In the materials and methods section, lines 317-319, you mentioned that you included 19 patients with BBIB and 13 with PTB in the study. What is the actual study sample?
- Description of statistical tests: You should describe which statistical tests you applied and how.
- Explanation for studying only vitamin D deficiency: It would be advisable to explain why you studied only vitamin D deficiency and not folic acid deficiency as well.
- P-value formatting: The P-value in Table 5 should be presented as a numerical value. For example, 8.1 x 10^-4 should be written as 0.00081.
Conclusions:
- The study includes only 19 women, which limits the statistical power and generalizability of the results.
- The results are not validated on an independent sample, which would increase confidence in the conclusions.
Author Response
Responses to the reviewers were included in the attached rebuttal.
